# Pollen Molecular Identification from a Long-Distance Migratory Insect, *Spodoptera exigua,* as Evidenced for Its Regional Pollination in Eastern Asia

**DOI:** 10.3390/ijms24087588

**Published:** 2023-04-20

**Authors:** Huiru Jia, Tengli Wang, Xiaokang Li, Shengyuan Zhao, Jianglong Guo, Dazhong Liu, Yongqiang Liu, Kongming Wu

**Affiliations:** 1State Key Laboratory for Biology of Plant Diseases and Insect Pests, Institute of Plant Protection, Chinese Academy of Agricultural Sciences, Beijing 100193, China; jhuiru@163.com (H.J.); 15978356769@163.com (T.W.); lixiaokang2016@163.com (X.L.); zhaosy90@126.com (S.Z.); jianglongguo88@163.com (J.G.); liudazhong94@163.com (D.L.); lyq364467268@163.com (Y.L.); 2Guangdong Laboratory for Lingnan Modern Agriculture, Guangzhou 510640, China; 3College of Plant Protection, Shenyang Agricultural University, Shenyang 110866, China

**Keywords:** pollen, *Spodoptera exigua*, migration, pollination, conspecific attraction

## Abstract

Understanding plant-insect interactions requires the uncovering of the host plant use of insect herbivores, but such information is scarce for most taxa, including nocturnal moth species, despite their vital role as herbivores and pollinators. In this study, we determined the plant species visited by an important moth species, *Spodoptera exigua*, by analyzing attached pollen on migratory individuals in Northeast China. Pollen grains were dislodged from 2334 *S. exigua* long-distance migrants captured between 2019 and 2021 on a small island in the center of the Bohai Strait, which serves as a seasonal migration pathway for this pest species, and 16.1% of the tested moths exhibited pollen contamination, primarily on the proboscis. Subsequently, 33 taxa from at least 23 plant families and 29 genera were identified using a combination of DNA barcoding and pollen morphology, primarily from the Angiosperm, Dicotyledoneae. Moreover, the sex, inter-annual, and seasonal differences in pollen adherence ratio and pollen taxa were revealed. Notably, compared to previously reported pollen types found on several other nocturnal moths, we found that almost all of the above 33 pollen taxa can be found in multiple nocturnal moth species, providing another important example of conspecific attraction. Additionally, we also discussed the indicative significance of the pollen present on the bodies of migratory individuals for determining their migratory route. Overall, by delineating the adult feeding and pollination behavior of *S. exigua*, we advanced our understanding of the interactions of the moths with their host plants, and its migration pattern, as well as facilitated the design of (area-wide) management strategies to preserve and optimize ecosystem services that they provide.

## 1. Introduction

Plants and insects across the globe have co-evolved into intricate patterns of interactions that are often mutually beneficial. For instance, adult insects visit flowers to obtain food, typically in the form of pollen or nectar for reproduction or flight; meanwhile, plants receive services rendered by pollinators, as they facilitate plant reproduction and, thus, support the majority of the world’s plant diversity [1,2,3]. Therefore, a comprehensive knowledge of the interaction between insect visitors and pollinators with flowering plants is essential for interpreting ecological and evolutionary phenomena, as well as maintaining ecosystem functioning and stability. While determining the range of host plants used by specific insect herbivores is the first and most critical task for addressing the above issues [4,5], such investigations have been lacking in most flower-visiting insects, especially in non-bee taxa, despite their significant role in the provision of pollination ecosystem services being increasingly reported in recent years [6,7].

Nocturnal moths are one often-neglected yet critically important group of non-bee pollinators that demand attention. As a crucial member of the largest Lepidoptera order, this insect group is probably the most diverse and widely distributed group of phytophagous insects on the planet, with over 40,000 described species [8,9]. Most moth species are well known as herbivorous, whose adults visit flowers to feed on nectar and/or pollen as their principal carbohydrate source and thereby aiding in plant pollination. Indeed, accumulating evidence has demonstrated that they are major pollinators of a diverse range of plant species, particularly those who visit flowers at night or in the early morning when bees are less active [10,11,12]. Moreover, many of them are serious agricultural and forestry pests with the ability to migrate over long distances, always causing significant damage to a variety of crops [13]. This further emphasizes the need for studies of feeding behavior in this group, which can provide scientific guidance for integrated pest management practices. However, such information has been limited to only a few species within this group, such as *Helicoverpa zea* [14,15] and several *Agrotis* species [16,17]; the remaining majority of species are at best uncertain, and at worst, remain unexplored.

Various techniques have been developed to explore insect feeding preferences and dietary profiles, including direct laboratory and field observations, stable isotopes, pollen analysis, and molecular gut-content analysis [18,19,20]. Among these methods, pollen analysis is a frequently used tool now, particularly for studying the above-mentioned nocturnal moths whose adult diets can be challenging to examine using traditional approaches, such as direct laboratory and field observations, due to their nocturnal or high-altitude habits. Of note, recent advances in pollen identification techniques, such as scanning electron microscopy (SEM) and DNA barcoding, have made it possible to identify pollen grains to the species or genus level [21], thereby further promoting the use of pollen analysis for studying the dietary composition of insect herbivores. In fact, this pollen analysis approach has been extensively and successfully employed in recent decades on a wide range of flower-visiting insects, particularly in multiple night-flying noctuid moths [16,17,22,23].

In order to gain a better understanding of the interactions of moth pollinators and plants, we utilized pollen analysis to investigate the host plant use of adults of the beet armyworm *Spodoptera exigua* (Hübner). *S. exigua* is an important migratory noctuid species with worldwide distribution, whose larvae attack over 90 plant species in Asia, including several major crops such as sugar beet, cotton, soybean, and potatoes [24,25,26]. In our study, we aimed to answer several questions: (i) Can *S. exigua* moths transport pollen over long distances? (ii) If so, what types of pollen species are transported by this moth species, and how do the frequency of pollen deposition and pollen taxa vary between sexes and over months and years? (iii) What are the characteristics of the host plants inferred from the pollen grains attached? (iv) Are there any plants with a highly limited distribution in a local area that can be used to assess long-distance movement? Our previous 14-year study of *S. exigua* performed by searchlight trapping on a small island, namely Beihuang (BH) in the center of the Bohai Strait, a seasonal migration pathway of many insect species in eastern Asia, showed that this species undertake regular long-distance migrations across the Bohai Strait every year [27]. Therefore, in the present study, we combined morphologically based methods through scanning electron microscopy (SEM), and molecular techniques based on DNA barcoding to identify pollen grains attached to long-distance migrants of the *S. exigua* moths captured at this isolated site (BH) during 2019–2021.

## 2. Results

### 2.1. Morphology of the Proboscis and Pollen Detection

Previous studies revealed that pollen grains attach primarily to the proboscis but occasionally to other body parts [14,16], so pollen examination focused on the proboscises of collected *S. exigua* moths. SEM observations revealed that the proboscis of this species is composed of two elongated, interlocked maxillary galeae that enclose the food canal by dorsal and ventral legulae. The surface of the food canal bears closely connected and smooth semicircular ridges, gradually tapering toward the proboscis tip (Figure 1B). Three types of sensilla are noticeable on the tube-like proboscis: sensilla trichodea, basiconica, and styloconica.

During the entire experiment (2019–2021), 2334 migratory *S. exigua* moth specimens, 907 females and 1427 males, were collected on BH for examination. A total of 404 moths (17.30%) were found to have pollen adhered to their proboscis. When pollen was found on the proboscis, many grains were typically present (Figure 1C,D), but only one or two grains were found on rare occasions. This suggests active contact through feeding rather than, for example, casual contact via wind-blown contamination. 

### 2.2. Plant Hosts Inferred from Pollen

From the attached pollen grains on 404 *S. exigua* moths, 33 distinct pollen taxa were identified based on external morphology, size, and shape (Figure 2 and Figure 3). These examined pollen species varied in shape from spherical to elliptic to triangular. In addition to the three pollen species (Plate 1–2, 15, Figure 2), all pollen grains were individual pollen grains called “monad”. Moreover, the size of these pollen grains varied greatly between plant species, ranging from very large grains (>200 µm) (Plate 1–2, Figure 2) to small size grains in members of the Asteraceae (10–25 µm) (Plate 33, Figure 2), whereas 81.82% of (27/33) pollen species were typically medium-sized or small, with diameters of around 26–50 µm and 10–25 µm, respectively. Generally, the surface features of pollen grains are the most important morphological character for the identification of pollen species. Six types of sculpturing patterns were observed here, including smooth (psilate), net-like (reticulate), or one looking like a ball of string (striate), and many pollen grains even had spine-like projections (echinate).

Meanwhile, pollen grains of each type were further confirmed using the ITS2 locus. The resulting molecular evidence having separated pollen grains beyond the plant genus level, 30 pollen taxa from 23 plant families and at least 29 genera were successfully identified, supported by as high as 99–100% similarity in sequence data and coverage. Three pollen taxa were unidentifiable due to a lack of descriptive morphological characters caused by field of view or lack of reference pollen (Plate 21, 25, 30, Figure 2). Regarding these identifiable pollen types, they pertained to the Asteraceae (6), Fabaceae (2) and Polygonaceae, (2) families, with many of them (20/33, 66.67%) being identified to species level: *Cynanchum acutum*, *Melia azedarach*, *Amorpha fruticosa*, *Flueggea suffruticosa*, *Solanum lycopersicum*, *Adenophora trachelioides*, *Persicaria orientalis*, *Gypsophila paniculata*, *Zanthoxylum schinifolium*, *Suaeda glauca*, *Cannabis sativa*, *Rubia cordifolia*, *Cuscuta japonica*, *Fagopyrum esculentum*, *Ambrosia trifida*, *Chenopodium album*, *Artemisia annua*, *Aster tataricus*, *Bidens alba*, *Chrysanthemum indicum*. A total of 10 samples were identified to the genus level: *Vincetoxicum* L., *Ligustrum* L., *Brassica* L., *Pinus* L., *Pithecellobium* L., *Festuca* L., *Oenothera* L., *Lonicera* L., *Asparagus* L., *Artemisia* L. Notably, most plant hosts were found from one moth, with only a few adults carrying pollen from different plant species.

### 2.3. Sex, Annual, and Seasonal Differences in Pollen Adherence Ratio

When years were grouped for analysis, there were significant sex-related differences in the annual mean frequency of pollen occurrence on *S. exigua* moths, with 15.1% (137/907) of females and 18.7% (267/1427) of male moths contaminated with plant pollen (Chi-square; χ^2^ = 5.0374, df = 1, *p* < 0.0248). However, when the years were analyzed individually, this significant difference was only observed in one year, 2020 (Chi-square; χ^2^ = 9.801, df = 1, *p* = 0.0017) (Figure 4A).

Similarly, the number of moths bearing pollen grains varied year-by-year. While the *S. exigua* moths trapped in summer (July–August) had a relatively higher pollen adherence ratio (18.5%, 182/982) than those trapped in autumn (September–October) (16.4%, 222/1352), the difference between the two seasons was not significant except in 2021 (2021: Chi-square; χ^2^ = 4.277, df = 1, *p* = 0.0386) (Figure 4B).

### 2.4. Annual and Seasonal Shifts in Pollen Taxa

There was a significant difference in the pollen species carried by *S. exigua* moths from year to year. During the study period (2019–2021), the greatest number of pollen taxa was found in individuals sampled in 2021, with 22 pollen species detected and 19 species detected in the other two years. A Venn Diagram shows that 16 of 33 pollen species were detected in just one year. Obviously, the greater the number of insects examined, the greater the diversity of pollen taxa detected (Figure 5A).

Additionally, seasonal variation in pollen species was also observed, with 19 species identified from summer migrants (July–August), and with *Cynanchum acutum*, *Amorpha fruticose*, and *Chenopodium giganteum* being the most common plant hosts (i.e., accounting for 52% of total carrier rate). Conversely, 15 pollen species were identified from autumn migrants. Among them, pollen of Asteraceae such as *Chrysanthemum zawadskii*, *Artemisia* L., and *Ambrosia trifida* were most commonly found on the moths, with >62.2% of the collection specimens contaminated with these pollens (Figure 5B).

### 2.5. Characteristics of Pollen-Bearing Plants

Further analysis of the characteristics of the pollen species showed that the plant species identified for the adherent pollen grains represented a variety of growth forms, including trees, shrubs, vines, and herbs. They were primarily angiosperms, with a few gymnosperms (Chi-square; χ^2^ = 346.67, df = 1, *p* < 0.001) or monocotyledons (Chi-square; χ^2^ = 562.99, df = 1, *p* < 0.001), and pollen species from herbaceous plants outnumbered those from woody plants (Chi-square; χ^2^ = 112.26, df = 1, *p* < 0.001). So, migratory moths of *S. exigua* most likely visited flowers of herbaceous angiosperms.

### 2.6. Diversity of Attached Pollen across Different Noctuid Moths

Our comparative analysis of the 33 pollen types identified in migrant *S. exigua* adults with those of other previously studied moth species revealed that pollen adhering to nectar-feeding moths show high similarity, with almost all identified pollen species in *S. exigua* moths being found on two or more other insect species (Figure 6). These pollen species can be divided into two distinct subgroups based on their distribution pattern: one group (Group A) consists of 15 pollen species such as *Cynanchum acutum* (Plate 1, Figure 2), *Vincetoxicum* L.(Plate 2, Figure 2) and *Pithecellobium* L.(Plate 15, Figure 2), which were only detected on moth species at an extremely high frequency, while the remaining 17 species constituted the other group (Group B) such as pollen of Asteraceae, *Chrysanthemum zawadskii*, *Artemisia* L., and *Ambrosia trifida*, which were equally common on the non-moth species. These results improved our understanding of the interactions of flowering plants and their herbivores.

## 3. Discussion

As insect adults frequently become contaminated with pollen during nutritional supplementation, identifying pollen found on an insect’s exterior provides a new means of determining the insect’s host plants. Indeed, attached pollen grains are commonly used as a natural marker to reveal foraging patterns or host plant associations for different types of pollinators [28,29]. In this study, we conducted serial investigations for pollen grains adhering to this species to reveal the host plant foraging range of Chinese migratory populations of the beet armyworm, *S. exigua*. For this investigation, we selected BH island as the sample collection site. BH island is a small island located within China’s Bohai Bay and is positioned within one prime (insect) migration corridor, and constitutes a unique setting to explore cross-regional insect pollination patterns over time. Notably, the trapping site (BH) has considerably facilitated the research on insect migration in eastern Asia. For instance, our research group at the site has elucidated the overseas migration behavior of numerous insect species across multiple orders, including our study species, the beet armyworm *S. exigua* [30,31]. During the entire experiment (2019–2021), 2334 *S. exigua* long-distance migrants were collected at BH, with catches of this species occurring only in 4 months from July to October, which was in accordance with previous migration observations for this species in the same location [27].

From the 2334 migratory *S. exigua* individuals captured in BH island, 16.1% of moths exhibited pollen contamination. The pollen detection rates of *S. exigua* were comparable to those observed in other noctuid moths collected in the same place, 17.03% for *A. segetum* [17], but lower than the 24.59% for *M. brassicae* [32] and 28.82% for *H. trifolii* moths [33]. As suggested in previous studies [16,34,35], the large difference in the frequency of pollen deposits among different insects can be ascribed to diverse elements, including plant phenology, nectar viscosity, pollen grain characteristics, migratory route, and antennae or mouthpart structure of insects, which can affect flower visitation patterns and associated variability in the frequency that an insect carries pollen. It should be noted that we may not have found all of the pollen-contaminated moths because some studies have proposed that long-distance migration greatly influences the number of moths possessing pollen, and that pollen detection rates significantly declined with distance from the feeding plants. For example, Forup and Memmott (2005) [36] reported that 82% of individual insects carried pollen from the plant species they had just visited in ancient and restored heathland in Southwest England. In contrast, Gibson et al. [37] (2006) found that 73% of visitors of arable weeds had pollen on their bodies. Additionally, although no significant differences existed among them, the higher number of moths possessing pollen in spring populations compared to late summer populations is most likely due to a greater number of flowering plants in the spring. Nonetheless, pollen found on *S. exigua* long-distance migrants provides direct evidence for the pollen-foraging behavior of this species, and migrating individuals can transport pollen over long distances.

At present, traditional morphologically based identification by SEM and DNA barcoding methods are the two most common methods for pollen identification. The microscopy-based method allows for direct viewing of attached pollen. It provides more detail for studying the surface morphology of pollen grains [38]. Still, it is time-consuming, requires significant expertise, has a low level of resolution, and may be unable to distinguish pollen from closely related plant species with similar morphology [39,40]. Alternatively, the latter molecular approach, which directly uses information from DNA sequences, can overcome the former approach’s difficulties and greatly facilitate pollen identification research. However, this method has significant limitations. For instance, it heavily relies on reference databases such as GenBank, making unambiguous, reliable identification of host plants that lack species reference sequences using molecular markers impossible [41]. As a result, similar to our previous pollen identification studies for other insects [16,21], we concurrently employed the two most common methods to identify pollen grains adhering to migratory *S. exigua* moths. This combination can compensate for the respective limitations of the two pollen identification methods, offering a more accurate, efficient, and comprehensive approach to identifying pollen grains with increased resolution and specificity, while saving time and reducing the dependence on expert knowledge.

Using the novel approach described above, we detected 33 plant species from 23 plant families and at least 29 genera in *S. exigua*, primarily Angiosperms in the Dicotyledoneae family, including herbs, shrubs, and vines. To the best of our knowledge, this is the first study to describe the host plant foraging range of Chinese populations of the beet armyworm, *S. exigua*. Aside from confirming the role of *S. exigua* as a key pollinator of wildflowers, our work identified its most important (seasonal) foraging resources, indicating that this species typically exhibit clear preferences for particular plant species. For example, summer migrants relied heavily on *C. acutum*, *A. fruticose*, and *C. giganteum,* whereas autumn migrants carried pollen from Asteraceae family plants more frequently. Some of these (preferred) host plant species may play a more important role in *S. exigua* population growth [42,43] and may warrant further attention in nutritional ecology studies. Meanwhile, using SEM, we obtained morphological characteristics of these examined pollen species, such as shape, size, and exine ornamentation patterns, which agreed well with their typical features. We are certain that all plant pollen-source samples were correctly identified by combining their morphological examination results with their molecular information.

Notably, when we compared the identified 33 pollen species to the identified pollen taxa adhering to seven reported moth species, whose adult feeding range was equally characterized using pollen-grain analysis method by our research team in the same study area (BH) [16,17,32,33] (i.e., Liu et al., 2016; Chang et al., 2018; He et al., 2022), we found that almost all pollen taxa shared one or more moth species, regardless of the number of tested moths used, suggesting that flowers pollinated by nocturnal moths may exhibit a degree of conservation. These compelling results provide—to our knowledge—the first evidence of moth species probably exhibiting conspecific attraction for habitat selection. The topic of conspecific attraction has been frequently reviewed within a taxa including insects [44]; for example, Schäpers et al. (2019) [45] reported that fruit flies and butterflies are more likely to pollinate plants within the same species or plant group, particularly when the plants are densely packed, have synchronized flowering periods, and are in high-quality environments. This may be partly explained by the findings of Guo et al. (2020) [46], who found that conserved ORs coevolved with the tube-like proboscis and maintained functional stability throughout the long-term coexistence of Lepidoptera with angiosperms. Additionally, some moth-specific plants were identified using one non-moth species as the outgroup, such as *Pithecellobium* spp., *C. acutum*, and *Oenothera* L., which were uniquely detected in more than two moths. The flowers of these (preferred) host plant species may contain specific attractant volatile components, making them potential targets for developing floral attractants for managing moth pests. This topic requires further investigation.

*S. exigua* has long been known as a long-distance migrant, and as mentioned above, its seasonal migration movements have been extensively studied on a small remote island located in the center of the Bohai Strait, in northern China, using searchlight trapping, ovarian dissection and genetic markers by our study group for many years [26,27]. However, the exact geographic origin of moths and their migration routes has still not been confirmed. The identification of pollen found on an insect’s exterior could be used to determine the insect’s migration route and geographical origin, in consideration of the fact that some plants only grow in specific ecological zones or geographical locations [28,47]. Indeed, pollen has been a natural marker in several migratory insects, including multiple moth species, over the past decades [33,48,49,50]. In our study, we also explored the indicative significance of the pollens presented on the bodies of migratory individuals for determining their migratory route. Similar to what has been found in other migratory moth species on this site [16,17,32,33], some pollen species from plants endemic to central and southern China, such as *M. azedarach* and *L. lucidum*, were observed on the proboscis of summer *S. exigua* migrants, suggesting that they likely foraged on these flowering plants and subsequently undertook a northward migration using the prevailing southwesterly winds across the Bohai sea in summer. This migration undoubtedly facilitates genetic exchange via pollen among plant populations across a large areas that is facilitated by adult *S. exigua*. Findings from this study will improve our knowledge on migratory behavior and regional outbreak of *S. exigua* in eastern Asia. Although our pollen analysis has yielded significant insights into the natal origins of *S. exigua* captured in BH during spring, further research should be needed for better understanding their trans-regional migration patterns by employing additional tracking methods such as radar observation [51], elemental tracing in insect bodies [52], trajectory analysis [53], and genetic diversity assessment [54] for the BH *S. exigua* population. This comprehensive approach will give us a deeper knowledge of the seasonal movement and pollination patterns of this species in Eastern Asia.

## 4. Materials and Methods

### 4.1. Moth Collection

The *S. exigua* moths used in this study were collected at the Beihuang (BH) field station of the Chinese Academy of Agricultural Sciences (Jinan, Shandong Province, China, 38°24′ N, 120°55′ E) using a vertical-pointing searchlight trap (model DK.Z.J1000B/t) equipped with a 1000 W metal halide-lamp (model JLZ1000BT; Shanghai Yaming Lighting Co., Ltd., Shanghai, China). Full details of this collection site (BH) are mentioned in our previous publications on insect migration [55,56].

From April 2019 to November 2021, sub-samples of 20 migratory *S. exigua* moths (or all individuals if the total capture size was below 20) were randomly taken from moths trapped each night, transferred to a 2 mL tube, and immediately stored in a freezer (−20 °C) until use.

### 4.2. Microscopic Examination of Pollen

Previous pollen studies revealed that pollen could adhere to various insect body parts (e.g., antennae, eyes, legs) but mainly to the proboscis [14,48,49]. Therefore, similar to our previous pollen analysis, special attention was paid to the proboscises of collected migratory *S. exigua* moths [17,32].

Light microscopy and scanning electron techniques were used for pollen examination and analysis, as reported by Liu et al. (2016) [16]. Briefly, the proboscis of individual moths was excised with non-magnetic tweezers, placed on a glass slide, and examined at 200× magnification with a stereomicroscope (Olympus SZX16, Pittsburgh, PA, USA) to determine the presence or absence of pollen on the insect. The suspected pollen grains were gently removed from the moth proboscis and mounted on aluminum stubs with double-sided sticky tape. Finally, pollen samples were sputter-coated with gold. They were visualized using a Hitachi S-8010 cold field emission scanning electron microscope (Hitachi, Tokyo, Japan) at the Electronic Microscopy Centre of the Institute of Food Science and Technology (CAAS, Beijing, China).

### 4.3. Molecular Analyses

To improve the accuracy of pollen identification, molecular tactics based on DNA barcodes, another efficient tool for recognizing pollen types, were also used to assess the pollen grains found on *S. exigua* moths. Genomic DNA was extracted from single pollen grains using protocols modified from a previous study [57]. Briefly, the pollen grain was placed in individual PCR tubes containing 5 µL of lysis solution (0.1 M NaOH plus 2% Tween-20) and incubated in a thermal cycler for 17 min 30 s at 95 °C (GeneAmp PCR System 9700, Applied Biosystems, Foster City, CA, USA). 5 µL of Tris-EDTA (TE) buffer was added to each lysis solution. The resulting solution was used as a template for subsequent PCR amplifications.

To improve species-level identification, two DNA barcoding loci for plants were used simultaneously, namely the *trnL* primers g and h to amplify the chloroplast *trnL* intron P6 loop [58] and the plant-specific primers ITS-p3 [59] and ITS4 [60] to amplify nuclear ribosomal Internal Transcribed Spacer region nrITS2. All partial regions were amplified separately in a 25 µL polymerase chain reaction (PCR) volume using the high-fidelity *Platinum* SuperFi II Master Mix (2X) (Invitrogen, Thermo Fisher Scientific, Waltham, MA, USA). The cycling parameters were 98 °C for 3 min, followed by 35 cycles of 98 °C for 1 min, 60 °C for 30 s, 72 °C for 1 min, and a final extension of 10 min at 72 °C. The expected sizes of the resulting PCR products were gel-purified using a Gel Extraction Kit (TransGen, Beijing, China) and then ligated directly into the pClone007 Vector (Tsingke BioTech, Beijing, China). A total of four positive clones were randomly selected, and Sanger sequenced with M13 primers at the Sangon Biotech Co., Ltd. (Shanghai, China) or Beijing Genomics Institute (BGI, Shenzhen, China).

### 4.4. Pollen and Plant Host Identification

Each pollen grain was identified based on its molecular and morphological characteristics and geographic distribution. Firstly, genetic sequences were compared with those in the National Center for Biotechnology Information (NCBI) database using the online BLASTn search program. If the sequence top bit score matched a single species, multiple species within a given genus, or multiple genera within a given family, the sequence was designated to that respective species, genus, or family. Sequences aligned with multiple families were termed ‘unidentifiable’ [61]. As a results, several sequence taxa were assigned to the rank of genus or family. Separate analyses were performed for each of the four tested markers, and the results were combined to identify specific pollen species. Identifications based on molecular data were further complemented by morphological characterization using published SEM images of pollen grains of Chinese flora [62,63] or online search engines and palynological databases (https://www.paldat.org/ (accessed on 10 February 2023)). Finally, we cross-checked the species-level identifications with the Flora of China Species Library (https://species.sciencereading.cn (accessed on 10 February 2023)) and the Plant Science Data Center (https://www.plantplus.cn/cn (accessed on 10 February 2023)) to determine the presence of plant hosts in the broader study area. By using this combination of techniques, we were able to accurately identify each pollen grain and determine the plant species visited by the nocturnal moths.

### 4.5. Data Analysis

To better understand “moth-host plant” interactions, we compared the identified pollen taxa found on migrant *S. exigua* adults to pollen types carried by several previously surveyed moth species. Seven moth species were selected as candidates: *Agrotis ipsilon* [16] (Liu et al., 2016), *Agrotis segetum* [17] (Chang et al., 2018), *Mythimna separate* [22], *Helicoverpa armigera* [23], *Mamestra brassicae* [32], *Hadula trifolii* [33], *Athetis lepigone* (unpublished data), and the hoverfly *Episyrphus balteatus* was used as a comparator species [21]. Our research group successfully elucidated the feeding behavior of the above eight species using pollen-grain analysis at the same study site (BH).

One-way analysis of variance (ANOVA) followed by Tukey’s test for multiple comparisons was used to evaluate differences in the frequency of pollen deposits on *S. exigua* moths sampled from different times. Wilcoxon rank-sum test was used to compare differences in the annual mean frequency of pollen occurrence on the female and male proboscises of *S. exigua* moths and differences in the annual mean frequency of pollen deposits on the proboscises of female, male, and total (female and male) *S. exigua* moths. A chi-square test was used to compare the differences in the rates of pollen deposits on female, male, and total (female and male) *S. exigua* moths each year and the characteristics of pollen source plants. All statistics analyses were performed in SPSS 13.0 (SPSS Inc., Chicago, IL, USA) [64].

## 5. Conclusions

In conclusion, identification of the host plant range of insect herbivores is critical in understanding plant-insect interactions. However, despite the importance of this group as herbivores and pollinators, as well as it being one of the most damaging pest groups to agriculture, such information is very limited for most taxa, such as nocturnal moth species. In this study, we determined the plant species visited by Chinese populations of the adult beet armyworm, *S. exigua*, in Eastern Asia using morphological analysis and DNA metabarcoding of pollen carried by the moths, and a wide host plant range (33 plant species from least 23 plant families and 29 genera) and extensive flower visitation network of *S. exigua* were determined. Moreover, a large-scale comparative analysis of our study further confirmed the phenomenon of conspecific attraction, which means that insects within the same taxon are more likely to pollinate plants within the same species or plant group. Some moth-specific plants were obtained, such as *Pithecellobium* spp., *C.acutum*, and *Oenothera* spp. Additionally, we also unveiled how *S. exigua* exhibits a seasonally adaptive migration pattern that spans northern and northeast China. These findings improved our understanding of moth-host plant interactions and provided invaluable guidance for (area-wide) management strategies. Although our study focuses on the role of nectar host plants in moth-plant interactions, it is worth noting that some of these nectar host plants may also serve as larval hosts. A comprehensive understanding of the relationships between larval and nectar hosts could contribute to a more in-depth knowledge of the ecology and conservation of these species. Future research on the moth-plant interaction could be helpful for getting a broader perspective on the dynamics of these mutualistic networks.

## Figures and Tables

**Figure 1 ijms-24-07588-f001:**
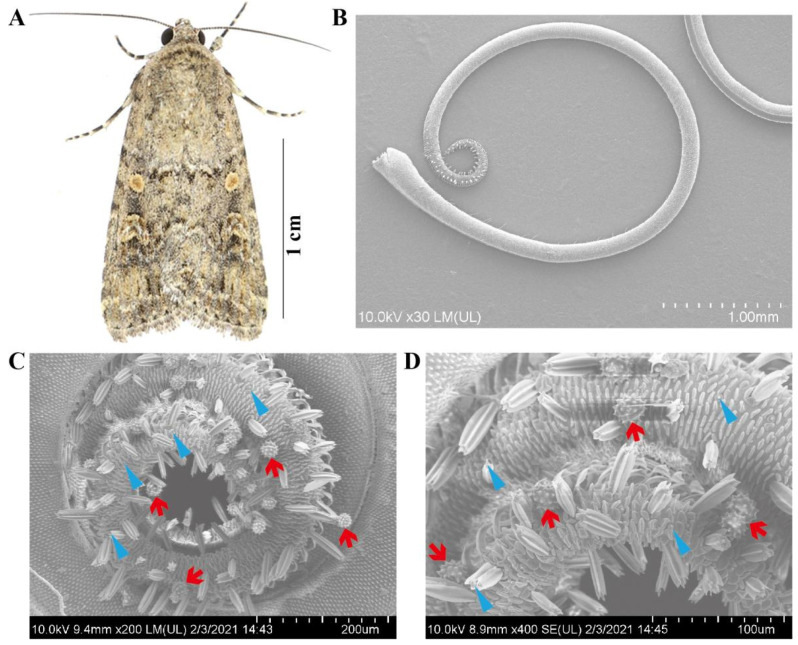
The beet armyworm, *Spodoptera exigua,* and the pollen adhering to the proboscis of this species. (**A**) Representative images of adult *S. exigua.* (**B**) Overall view of the adult proboscis. (**C**,**D**) Proboscis with attached pollen grains from species of Asteraceae, probably a representative of the tribe Heliantheae. The red arrow denotes the attached pollen grains on the proboscis, and the blue arrow indicates the sensilla.

**Figure 2 ijms-24-07588-f002:**
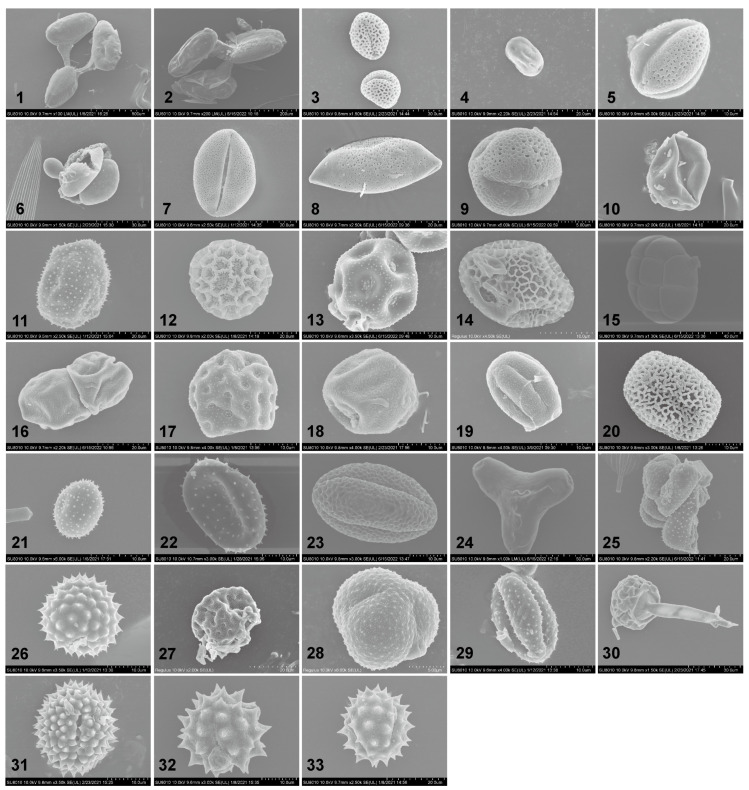
SEM images of the examined pollen species found adhering to *Spodoptera exigua* moths. 1. *Cynanchum acutum*; 2. *Vincetoxicum* L.; 3. *Ligustrum* L.; 4. *Melia azedarach*; 5. *Brassica rapa*/*Brassica napus*/*Brassica oleracea*/*Brassica juncea*; 6. *Pinus* L.; 7. *Amorpha fruticosa*; 8. *Asparagus* L.; 9. *Flueggea suffruticosa*; 10. *Solanum lycopersicum*; 11. *Adenophora trachelioides*; 12. *Persicaria orientalis*; 13. *Gypsophila paniculata*; 14. *Zanthoxylum schinifolium*; 15. *Pithecellobium* L.; 16. *Festuca* L.; 17. *Suaeda glauca*; 18. *Cannabis sativa*; 19. *Rubia cordifolia*; 20. *Cuscuta japonica*; 21. unknown; 22. *Lonicera* L.; 23. *Fagopyrum esculentum*; 24. *Oenothera* L.; 25. unknown; 26. *Ambrosia trifida*; 27. *Chenopodium album*; 28. *Artemisia annua*; 29. *Artemisia* L.; 30. unknown; 31. *Aster tataricus*; 32. *Chrysanthemum indicum*; 33. *Bidens alba*.

**Figure 3 ijms-24-07588-f003:**
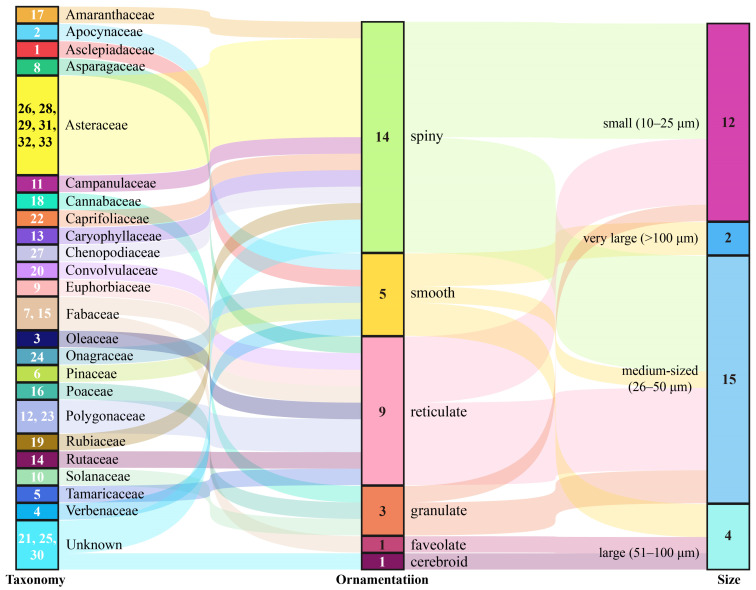
Sankey plots for the taxonomic information and morphological features of the examined pollen species. The number inside the Taxonomy bars represent the pollen type in Figure 2, while the numbers within the ornamentation and size bars denote sample sizes.

**Figure 4 ijms-24-07588-f004:**
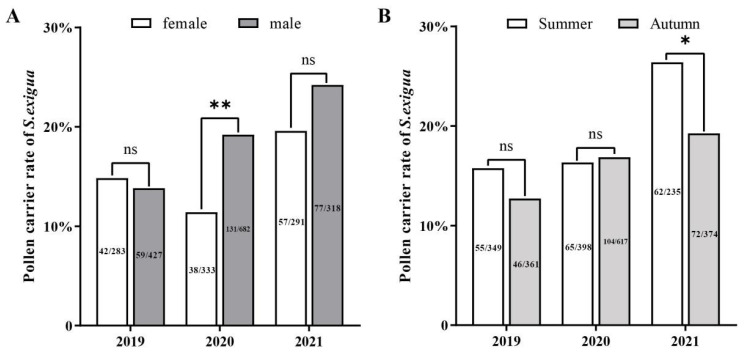
Frequencies of pollen deposition on migratory individuals of *Spodoptera exigua* among male and female moths (**A**) and different migration seasons (**B**) during 2019–2021. The numbers on the bar plots represent the number of adults with pollen/number of adults examined. * and ** stand for significant different at *p* < 0.05 and *p* < 0.01), respectively; ns: stand for no significant different.

**Figure 5 ijms-24-07588-f005:**
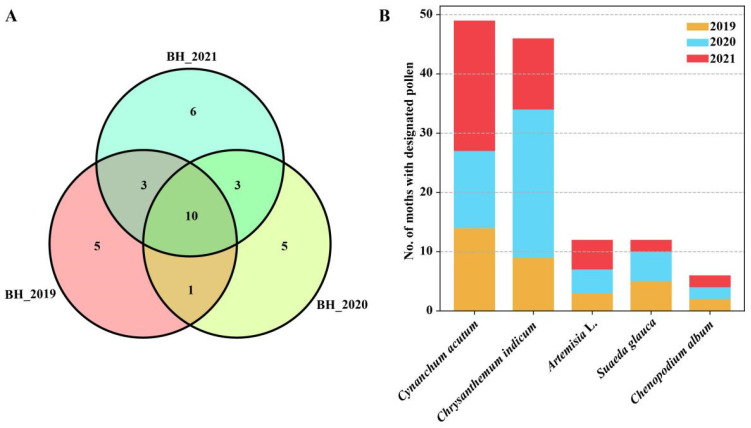
Annual and seasonal shifts in the pollen types found on migrant *Spodoptera exigua* adults sampled from Beihuang island (Bohai sea) during 2019–2021. (**A**) Venn diagram showing the common and uniquely attached pollen species in three years. The numbers in the Venn diagram indicate the number of pollen taxa identified. (**B**) Bar plot showing the number of moths contaminated with the most common pollen taxa during 2019–2021.

**Figure 6 ijms-24-07588-f006:**
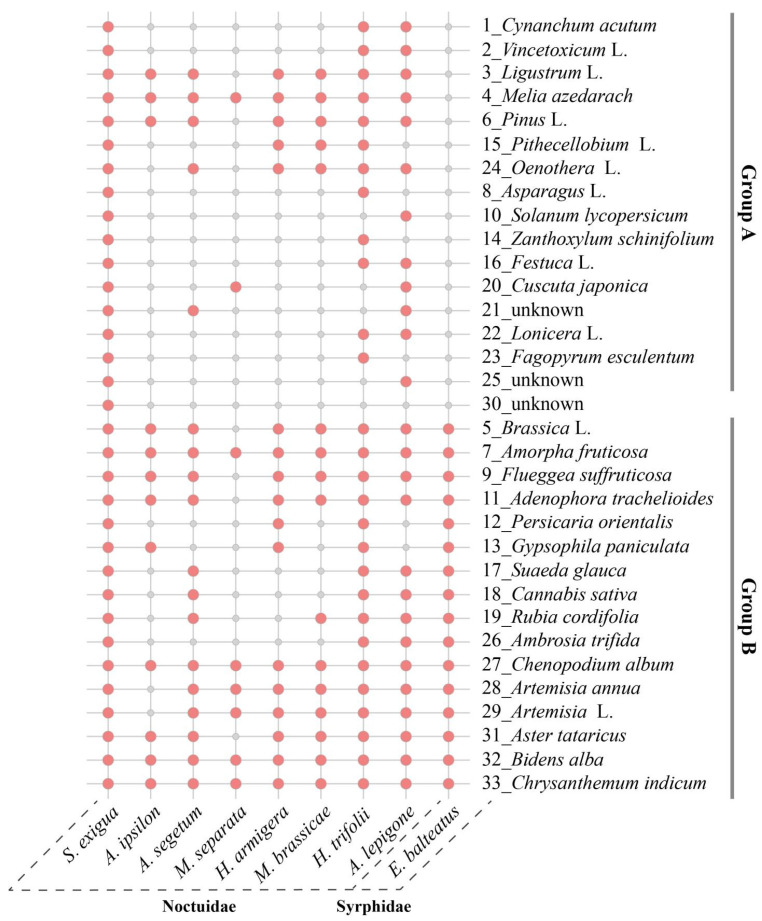
Comparative analysis of the attached pollen species on migrant *Spodoptera exigua* adults with those of detected pollen taxa in other insects sampled in the same study site, Beihuang island (Bohai sea). Red bubbles indicate the presence of designated species in given insect species.

## Data Availability

Data is contained within the article.

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
