# Peer review of "Pollen Molecular Identification from a Long-Distance Migratory Insect, Spodoptera exigua, as Evidenced for Its Regional Pollination in Eastern Asia"

_ijms, 2023, doi:10.3390/ijms24087588_

Round 1
Reviewer 1 Report
The work by Huiru Jia and collaborators evaluates the role of Spodoptera exigua adults as pollinators in an island of China. The novelty of the work is the authors confirmed the assemblage of plant species visited as nectar sources by migrant adults of S. exigua through a combination of molecular and palynological analyses of the pollen grains carried in the probosces of the 2,334 captured moths between 2019 and 2021. The authors identified 33 different pollen types carried by S. exigua individuals an compared these findings with the pollen carried by other seven noctuid moths and a syrphid fly previously studied by the group in the area.
I consider that the work is an interesting contribution to the journal but several changes need to be done before its final publication.
In the title is highlighted that S. exigua is a long-distance migratory insect, and is also mentioned in several parts of the manuscript that migration of this moth species was previously studied by the research group. I consider it will be important to give more details about the migration of this species, such as possible routes, seasonality, and possible distances that moths migrate. In the abstract the authors mentioned “we also discussed the indicative significance of the pollen present on the bodies of migratory individuals for determining their migratory route”, but I am afraid I did not find any information about this topic in the discussion section.
There are several words that are used throughout the manuscript that I consider are incorrectly used, and should be changed by other words or clarified to avoid confusions.
Host plants: “host plants” is frequently used for caterpillars, I suggest to explain at the first mention that the authors refer to the plant species used as nectar sources by adult moths.
Coevolution: I consider the experimental design of the work does not allow to confirm coevolution between S. exigua and the different pollen types carried by captured moths. Coevolution is the evolution involving successive changes in two or more ecologically interdependent species that affect their interactions. To confirm coevolution is necessary to apply comparative methods taking into account relatedness among both, the guild of moth-pollinated plants and the assemblage of moth species. I consider the design of the present work only inform about the plant-pollinator interactions among moths and the plants they visit as nectar sources. I suggest omitting the word coevolution throughout the manuscript. For an example of coevolution among noctuid moths and nightshades plant species see Ibañez et al. 2019 (https://www.sciencedirect.com/science/article/pii/S1055790318303828)
Contamination: Captured moths carried different types of pollen. It is very important that the authors highlight in the results and discussion sections that several pollen types belong to plant species that show floral traits adapted to nocturnal pollination by noctuid moths (e.g. nocturnal anthesis, pale corollas, nocturnal fragrance emission and nectar production) such as Oenothera, Pithecellobium, etc. Other species, such as the Asteraceae, are probably visited because they are the only nectar source available in the area or in certain time of the year they are very abundant. In these two cases moths probably act as efficient pollinators. However, it is necessary to clarify that several pollen types that belong to wind-pollinated species (such as Cannabis, Pinus of Festuca) are clearly contaminations. Thus, I suggest in the results to detail which of the pollen types belong to plants specifically pollinated by the moths and which are contaminations.
Preference: When authors mention about moth preferences I consider that they may feed in those plant species because they are the only nectar source in bloom or are very abundant in comparison of other species (e.g. asteracea species).
Results presentation:
Do the authors consider presenting the interactions among plants and moths (excluding those plant species that are clearly contamination such as wind-pollinated species mentioned above) in a bipartite network? In the pollination literature is frequently used the bipartite package of R Core Team to represent plant-pollinator interactions.
Figure 6. Syrphidae and Noctuidae subheadings are in the wrong position. Do not use Clade. Clade is a group of biological taxa that includes all descendants of one common ancestor and this is not the case. Which criteria the authors used to separate these two groups (clades a y b)?
Discussion: There is a another effective, cheap and less consuming time method to identify pollen grains carried by nocturnal pollinators, using semi-permanent slides under a light microscope (e.g. Sazatornil et al. 2016 https://besjournals.onlinelibrary.wiley.com/doi/full/10.1111/1365-2656.12509) that the authors could mention in the discussion.
Lines 300-306. I suggest attenuating this affirmation.
Author Response
Reviewer 1 #
The work by Huiru Jia and collaborators evaluates the role of Spodoptera exigua adults as pollinators in an island of China. The novelty of the work is the authors confirmed the assemblage of plant species visited as nectar sources by migrant adults of S. exigua through a combination of molecular and palynological analyses of the pollen grains carried in the probosces of the 2,334 captured moths between 2019 and 2021. The authors identified 33 different pollen types carried by S. exigua individuals an compared these findings with the pollen carried by other seven noctuid moths and a syrphid fly previously studied by the group in the area. I consider that the work is an interesting contribution to the journal but several changes need to be done before its final publication.
Response: We are grateful for your positive comments on our work and recognition of the novelty of our study. We have carefully addressed all the points and made necessary revisions accordingly. Please find our point-by-point response below:
- In the title is highlighted that S. exigua is a long-distance migratory insect, and is also mentioned in several parts of the manuscript that migration of this moth species was previously studied by the research group. I consider it will be important to give more details about the migration of this species, such as possible routes, seasonality, and possible distances that moths migrate.
Response: Accepted. Please find the changes in the following sections:
In the Introduction, we have expanded the discussion on the migration behaviour of S. exigua (Line 91-94).
In the Discussion section, we have further elaborated on the implications of these migration patterns for the role of S. exigua as pollinators (Line327-351).
We believe that these additions provide a more comprehensive picture of its migration and enhance the overall understanding of their role as pollinators. We hope these revisions address your concerns and improve the quality of our manuscript.
- In the abstract the authors mentioned “we also discussed the indicative significance of the pollen present on the bodies of migratory individuals for determining their migratory route”, but I am afraid I did not find any information about this topic in the discussion section.
Response: In light of your feedback, we have revised and expanded upon this point in our updated discussion section to ensure a more thorough and clear explanation (Lines 327-353).
- There are several words that are used throughout the manuscript that I consider are incorrectly used, and should be changed by other words or clarified to avoid confusions. I: Host plants: “host plants” is frequently used for caterpillars, I suggest to explain at the first mention that the authors refer to the plant species used as nectar sources by adult moths.II: Coevolution: I consider the experimental design of the work does not allow to confirm coevolution between S. exigua and the different pollen types carried by captured moths. Coevolution is the evolution involving successive changes in two or more ecologically interdependent species that affect their interactions. To confirm coevolution is necessary to apply comparative methods taking into account relatedness among both, the guild of moth-pollinated plants and the assemblage of moth species. I consider the design of the present work only inform about the plant-pollinator interactions among moths and the plants they visit as nectar sources. I suggest omitting the word coevolution throughout the manuscript. III: Contamination: Captured moths carried different types of pollen. It is very important that the authors highlight in the results and discussion sections that several pollen types belong to plant species that show floral traits adapted to nocturnal pollination by noctuid moths (e.g. nocturnal anthesis, pale corollas, nocturnal fragrance emission and nectar production) such as Oenothera, Pithecellobium, etc. Other species, such as the Asteraceae, are probably visited because they are the only nectar source available in the area or in certain time of the year they are very abundant. In these two cases moths probably act as efficient pollinators. However, it is necessary to clarify that several pollen types that belong to wind-pollinated species (such as Cannabis, Pinus of Festuca) are clearly contaminations. Thus, I suggest in the results to detail which of the pollen types belong to plants specifically pollinated by the moths and which are contaminations. IV: Preference: When authors mention about moth preferences I consider that they may feed in those plant species because they are the only nectar source in bloom or are very abundant in comparison of other species (e.g. asteracea species).
Response: Accepted. We have carefully revised the manuscript to address your concerns and ensure that the correct terms are used to convey our findings clearly and accurately.
Host plants: We have clarified in the first mention of "host plants" that we are referring to plant species used as nectar sources by adult moths.
Coevolution: You are correct that our study does not provide sufficient evidence to confirm coevolution between S. exigua and the different pollen types carried by the moths. We have removed the term "coevolution" throughout the manuscript and focused on describing the plant-pollinator interactions among moths and the plants they visit as nectar sources.
Contamination: Thank you for your insightful comments regarding the different types of pollen carried by the captured moths. In our study, when pollen was found on the proboscis, many grains were typically present, which suggested active contact through feeding rather than casual contact through wind-blown contamination. Although these plants rely on wind pollination, it does not exclude the possibility that insects, including moths, can feed on them. In fact, our previously published feeding experiments have demonstrated that moths can feed on the pollen of these types of plants (Jia et al., 2022).
Preference: Thank you for raising the point about moth preferences potentially being influenced by the availability and abundance of certain plant species, such as those with longer blooming periods or higher abundance (e.g., Asteraceae species). We agree that this could be a contributing factor in the observed moth preferences. As such, we have revised our discussion of moth preferences to acknowledge that moths may feed on particular plant species because they are the only nectar source in bloom or are more abundant compared to other species (e.g., Asteraceae species).
- Results presentation:
(1) Do the authors consider presenting the interactions among plants and moths (excluding those plant species that are clearly contamination such as wind-pollinated species mentioned above) in a bipartite network? In the pollination literature is frequently used the bipartite package of R Core Team to represent plant-pollinator interactions.
Response: Thank you for your suggestion to consider presenting the interactions among plants and moths in a bipartite network using the bipartite package of R Core Team, as frequently seen in the pollination literature. We appreciate your input and understand the potential benefits of using such a representation.
Initially, we did explore the possibility of using a bipartite network to visualize the interactions among plants and moths, as recommended (see following figure). However, we encountered some challenges, and the resulting network appeared chaotic and did not effectively convey our results in a clear and comprehensible manner. Therefore, we decided to maintain our current representation, which we believe better communicates our findings.
Additionally, we have taken into account your following comments on our original figures and have made revisions to improve their clarity and presentation. We hope that these adjustments address your concerns and provide a more accurate and accessible representation of our results.
Once again, we appreciate your valuable feedback and believe that our response addresses the concerns raised.
(2) Figure 6. Syrphidae and Noctuidae subheadings are in the wrong position. Do not use Clade. Clade is a group of biological taxa that includes all descendants of one common ancestor and this is not the case. Which criteria the authors used to separate these two groups (clades a y b)?
Response: We apologize for the confusion and have made the following corrections:
- a) We have fixed the position of the Syrphidae and Noctuidae subheadings to accurately reflect their respective groups.
- b) We have removed the term "clade" and replaced it with "group" to avoid confusion, as you correctly noted that a clade represents all descendants of one common ancestor.
- c) We have included a brief explanation of the criteria used to separate the two groups (a and b) in the Results section.
- Discussion:
(1) There is a another effective, cheap and less consuming time method to identify pollen grains carried by nocturnal pollinators, using semi-permanent slides under a light microscope that the authors could mention in the discussion (e.g. Sazatornil et al. 2016 https://besjournals.onlinelibrary.wiley.com/doi/full/10.1111/1365-2656.12509).
Response: Thank you for your suggestion to consider mentioning the alternative method of using semi-permanent slides under a light microscope for identifying pollen grains carried by nocturnal pollinators, as described in the study by Sazatornil et al. (2016). We have carefully reviewed the provided reference, and we found that the main focus of the study is on the role of specialized pollination by nocturnal hawkmoths in shaping plant-pollinator networks, rather than detailing specific pollen identification methods.
Given the absence of a detailed pollen identification method in the suggested reference, we have decided not to include this study in the discussion section of our manuscript. However, we appreciate your input and remain open to considering other relevant references that might enhance our understanding of pollen identification techniques in the context of nocturnal pollinators, should you have any further recommendations.
(2) Lines 300-306. I suggest attenuating this affirmation.
Response: Accepted. We have revised the statement to present a more cautious interpretation of our findings. The updated statement now reads:
"Notably, when we compared the identified 33 pollen species to the identified pollen taxa adhering to seven reported moth species whose adult feeding range was equally characterized using pollen-grain analysis method by our research group in the same study area (BH) [16-17, 32-33] (i.e. Liu et al., 2016; Chang et al., 2018; He et al., 2022), we found that almost all pollen taxa were shared with one or more moth species, regardless of the number of tested moths used, suggesting that flowers pollinated by nocturnal moths may exhibit a degree of conservation."
Reviewer 2 Report
This is a very innovative and polished piece of research, and I support everything the authors write about the relevance of the work. I only suggest adding a short paragraph distinguishing larval and nectar hosts. It is not critical to link pollen data to larval hosts; I assume at least some of the nectar hosts are also larval hosts. The importance of this research is not necessary to identify larval hosts. In a few sentences, they seemed to imply an association with larval hosts, but not explicitly. I would find it interesting to know if, in fact, nectar and larval hosts were different. This is why I noted the introduction could be improved, but it would be something raised in the introduction and addressed again briefly in the discussion.
Author Response
Reviewer 2
Comments and Suggestions for Authors
This is a very innovative and polished piece of research, and I support everything the authors write about the relevance of the work. I only suggest adding a short paragraph distinguishing larval and nectar hosts. It is not critical to link pollen data to larval hosts; I assume at least some of the nectar hosts are also larval hosts. The importance of this research is not necessary to identify larval hosts. In a few sentences, they seemed to imply an association with larval hosts, but not explicitly. I would find it interesting to know if, in fact, nectar and larval hosts were different. This is why I noted the introduction could be improved, but it would be something raised in the introduction and addressed again briefly in the discussion.
Response: Thank you for your kind words and support regarding the relevance and innovation of our research. We appreciate your suggestion to include a short paragraph distinguishing larval and nectar hosts and addressing the potential association between them.
- In response to your feedback, we have revised the introduction to provide a clearer distinction between larval and nectar hosts, and to emphasize the focus of our study on the nectar host interactions:
Line55-59:" Most moths species are herbivorous, whose adults visit flowers to feed on nectar and/or pollen as their principal carbohydrate source and thereby aiding in plant pollination. Indeed, accumulating evidence demonstrated that they are major pollinators of a diverse range of plant species, particularly those that flower at night or in the early morning when bees are less active [10-12]. "
Line80-84: “ In order to gain a better understanding of the interactions of moth pollinators and plants, we utilized pollen analysis to investigate the host plant use of adults of the beet armyworm Spodoptera exigua (Hübner). S. exigua is an important migratory noctuid species with world distribution, whose larvae attacks over 90 plant species in Asia, including several major crops such as sugar beet, cotton, soybean, and potatoes [24-26]. ”
- b) Additionally, we have included a brief mention of this topic in the discussion section to further address your query(Line457-463):
"Although our study focuses on the role of nectar host plants in moth-plant interactions, it is worth noting that some of these nectar host plants may also serve as larval hosts. A comprehensive understanding of the relationships between larval and nectar hosts could contribute to a more in-depth knowledge of the ecology and conservation of these species. Future research on the moth-plant interaction could be helpful for getting a broader perspective on the dynamics of these mutualistic networks."
We really hope that these additions address your concerns and provide a more comprehensive understanding of the various plant interactions involved in the life cycle of S. exigua. We appreciate your valuable feedback and look forward to any further comments or suggestions you may have.
Round 2
Reviewer 1 Report
Authors have carefully replied to my concerns and I am happy with the changes made.
However, I would like to confirm that Sazatornil et al. 2016 used the method I mentioned in my review to identify the pollen types carried by all the hawkmoths captured in several communities as the authors made in their present manuscript. The original cite for this method is Kislev et al 1972 and is already cited in the work of Sazatornil et al. 2016:
KISLEV, M., KRAVIZ, Z., & LORCH, J. (1972). Department of Botany, The Hebrer University of Jerusalem. IsRAEL JOURNAL OF BOTANY, 21, 57-75.
and the pdf can be downloaded from this researchgate link:
https://www.researchgate.net/profile/Mordechai-Kislev-2/publication/282764629_A_study_of_hawkmoth_pollination_a_palynological_analysis_of_the_proboscis/links/561baff508ae78721fa100b4/A-study-of-hawkmoth-pollination-a-palynological-analysis-of-the-proboscis.pdf
In our lab we found out this method of Kislev et al (1972) to be more efficient (considering both time and money) to identify pollen loads in comparison to the new molecular alternatives.